# Intercultural Opportunities to Evoke Empathy toward Minority Citizens: Online Contact with Chinese International Students at a Japanese Women's University

Chie Sugino

Department of Culture and Tourism, Komazawa Women's University, Tokyo 206-8511, Japan;
c-sugino@komajo.ac.jp

**Abstract:** Exchanges with Chinese students in Japan, who comprise the majority of international students, may be a worthwhile intercultural experience for Japanese students. However, because of the lack of contact between Chinese and Japanese students on campus, many Japanese students tend to form impressions of China through the media. This study aims to explore the factors influencing Japanese students' positive attitudes toward Chinese students and the former's awareness of stereotypes based on an online interview in Japanese conducted during an elective political science course. This study adopts a qualitative research methodology to analyze students' written reflections contextually. The content analysis revealed that Chinese students' "Japaneseness", characterized by Japanese language fluency, affected the perceptions of Japanese students, who changed their attitudes toward Chinese people and intercultural exchanges. Further discussion is necessary to determine whether Japanese students' preferences for "Japanese-like" international students may create new stereotypes rather than enhance diversity benefits. The findings illustrate the need for future research and practice on how students with no experience in communicating with minority citizens can overcome ethnocentric perspectives and embrace diversity.

**Keywords:** stereotypes; global citizenship; interculturalism; contact theory; qualitative content analysis; online learning; higher education; Japan; COVID-19



## 1. Introduction

China is one of Japan's closest neighbors and constitutes the largest group of foreign residents in Japan [1]. However, a recent government survey revealed that approximately 80% of Japanese respondents did not have an affinity toward China, a trend that has been observed throughout previous decades, reflecting deteriorating bilateral relations [2]. Although the largest group of international students in Japan is Chinese (47.1%) [3], contact between Japanese and Chinese students on campus remains limited [4], and only a few studies have been conducted on the effects of exchanges with Chinese students on Japanese students' attitudes toward Chinese people [5–7]. Intergroup contact could improve relations between in-group and out-group members [8–25], and the author's practice confirmed that Japanese students had favorable perceptions of Chinese students after viewing an online interview. Therefore, this study explored the reasons for Japanese students' positive attitudes toward Chinese students and their awareness of stereotypes after watching the online interview with two Chinese students. The social background of the study is provided, followed by an overview of the contact theory hypothesis and research on the effects of intercultural contact on Japanese students. The second section describes the context of the study, details of the online interview, and the data collection and analysis. The results are discussed considering the current educational environment in Japan.

### 1.1. Increasing Need for Intercultural Education in Japan

As Japanese society becomes more multicultural and multinational, stereotypes and prejudices about foreign nationals and people with foreign roots have gained attention [26]. Discrimination and hate speech against Korean Japanese have continued even after the implementation of the Hate Speech Act of 2016, which prohibits the public incitement of hatred such as malicious slander and expulsion against specific races, ethnicities, and nationalities [27]. Unconscious bias has also been observed at the forefront of the hospitality industry. A hotel receiving Olympics-related guests was criticized for posting signs saying "Japanese only" on the elevators [28]. A travel advertisement was framed to describe a destination as "attractive without inbound foreigners" because of the pandemic [29].

These examples show that Japanese people's "invisible cultural norms" and "social privileges that their rules are given tacit precedence (over the minorities')" are taken for granted by the majority of Japanese [30] (pp. 84–85). As for the youth, an international comparative survey revealed that the percentage of Japanese youth who thought "the citizens of their own country are equipped with cross-cultural understanding and readiness needed to fulfill their role as members of the global community" was the smallest among seven countries (France, Germany, Japan, Korea, Sweden, the UK, and the U.S.) [31]. The increasing number of out-group members, such as foreign residents and tourists, is insufficient to change Japanese people's attitudes toward them; conscious and positive interactions are essential [32]. Thus, intercultural education is necessary to encourage a change in Japanese attitudes [26,30].

### 1.2. Effects of Intergroup Contact

This study drew on Allport's contact theory hypothesis, suggesting that intergroup contact can reduce prejudice and improve relations between in-group and out-group members under certain conditions such as having equal group status, and common goals and support from authority [8]. Pettigrew emphasized "friendship potential" as a beneficial condition for favorable intergroup contact [9] (p. 173). According to Miller, personalized interactions (e.g., the self-disclosure of intimate information) promote trust and empathy toward other group members [10]. Reading online comments can be a means of online direct contact; those who read comments by out-group members had more positive emotions than those who were given the same general information [11]. While having a common in-group identity also helps reduce mutual bias [12], contact effects are strong when out-group members are seen as typical representations of the group [13].

While the traditional focus of contact theory has been on positive effects, which tend to happen more often than negative effects, recent research indicates that a person's past experiences may be relevant for negative contact effects [14]. Prior positive contact experiences may mediate the effects of the present negative contact [15].

Contact experiences need not be direct. Different forms of indirect contact, which has the advantage of not requiring the presence of these groups, have been researched [16]: extended contact [17]; vicarious contact [18]; parasocial contact [19]; imagined contact [20]; and e-contact [21].

More positive intergroup attitudes are engendered when a person knows an in-group member who has a positive relationship with an out-group member [17]. First-year German university students who observed interactions between Chinese and German students became favorably inclined toward direct contact with Chinese students [18]. Parasocial contact considers mass-media-mediated interaction as interpersonal contact, which reduces the prejudices of a majority group toward minority groups [19]. Online text chat as e-contact can reduce transgender prejudices [21]. For young children who do not have out-group contact experiences, imagining being a disabled child may help change attitudes toward people with disabilities [20].

Finally, the significant influence of languages used in contact needs to be noted. It has been pointed out that the language used in contact influences attitudes toward out-group

members; native speakers of a language felt positive about non-native speakers' use of the language during the contact [22] and lowered the "social borders" to others [23] (p. 676).

Although a body of research has been developed on contact theory, Paluck et al., noticing an increase in "light touch interventions" that are easy to implement in a brief period, have questioned the validity of the effects of these "subtle treatments" [24] (p. 550). Similarly, convenient sample collection and "self-reported outcomes" in experiments based on contact theory have been criticized [25] (p. 646). A large-scale randomized controlled experiment is suitable to observe the general effect of contact on reducing prejudices against other groups. However, contact theory can also provide a framework to understand individual experiences. Based on contact theory, practitioners can gain valuable insights reflecting intercultural activities in class.

*1.3. Japanese Students' Exchange Opportunities with International Students*

The number of international students in Japan has increased in recent years, although there has been a drastic decline owing to the pandemic [3]. However, accepting international students does not automatically increase contact with them [4–7]. Thus, based on the contact hypothesis, some studies carried out "educational intervention" to provide exchange opportunities between Japanese and international students [5–7]. Japanese students, who heard about the difficulties of living in Japan by interviewing international students and foreign residents as an assignment, had empathy for international students [5], developed a "deepened intercultural understanding by looking at Japan in relation to foreign culture" [7] (p. 84), and became conscious about "Japanese people and society's prejudices and arrogance to minorities" and stereotypes [5] (p. 136) [7]. Moreover, collaborative work with international students positively impacts attitudes toward the multicultural understanding of international and Japanese students [6].

While the effects of indirect contact and online intergroup contact on students have been explored overseas, previous studies in Japan were based on face-to-face interactions due to the lack of such opportunities on campus. However, the pandemic brought a drastic shift in online education. Furthermore, considering people's increasing use of SNS and prejudices often found online, exploring online contact is necessary. In particular, it needs to be investigated whether and how online contact influences students' perceptions of others.

*1.4. The Objective of the Study*

Based on the literature, the author considered Japanese students watching an interview of international students as intergroup contact and thus conducted an online interview with two Chinese students in Japanese. This study examined the overall positive impact of contact on Japanese students by analyzing their post-interview comments to answer the following research questions:

1.　What influenced students' favorable perceptions of Chinese students after the online interview?
2.　Did Japanese students become aware of the stereotypes? If so, what did they notice?

**2. Materials and Methods**

This study adopted qualitative research methodologies suitable for dealing with descriptive data and analyzing them contextually and inductively [33]. The qualitative approach was appropriate for exploring student perceptions using their written reflections as the data source [34,35].

*2.1. Context of the Study*

This study was conducted during an elective political science course at a women's university in suburban Tokyo, Japan, where women's universities still amount to 10% of all universities [36]. The course introduced issues with multicultural societies, immigration, and intercultural communication intended for, but not limited to, first-year students of humanities majors. The course used discussion sessions with small groups of students from

different years and majors to promote intercultural exchange in the classroom. However, the online learning that launched during the pandemic in 2020 made it difficult to have group discussions involving all students due to technical and preferential reasons. Instead, an online interview with international students was introduced in the 6th class of the course as a new attempt at intercultural experiences. Students from the 2021 class watched the recording of the 2020 interview as part of an asynchronous class. Students in both years submitted their reflections after watching the interview, which were then analyzed.

### 2.2. Participants

The participants were 132 students out of 158 enrolled in the above-mentioned course: 61 were from an online synchronous class in 2020 and 71 were from an asynchronous class in 2021. The purpose of the study was explained to them based on the university's code of research ethics and guidelines, and all participants consented to join. Over 70% of the participants were first-year students whose majors could include Japanese culture, English language and culture, communication, culture and tourism, psychology, and architecture and design, and were to be decided at the end of the semester. In previous lectures, the students learned the basics of intercultural communication, focusing on cultural differences, stereotypes, ethnocentrism, and cultural relativism.

The author explained the purpose of the class to all international students enrolled in the class, and invited them to join. Two out of three Chinese students agreed to participate in this study. They were first-year students from mainland China who joined the university after studying Japanese in language schools in Japan for about a year.

### 2.3. Proceedings of the 6th Class

One week before the interview, students in the 2020 class were asked to submit questions to Chinese students using Google Forms. This was voluntary, and the response rate was 53%. These questions were shared with two Chinese students (the interviewees) prior to class so that they could expect what was to be asked and exclude sensitive topics. They were open to any issues but wished to participate with their cameras off. The online interview was recorded and displayed in the 2021 class that was conducted in an asynchronous online setting.

To begin the interview, the author (interviewer) expressed appreciation of the Chinese students' willingness to answer any questions and reassured them that they did not need to answer sensitive or private issues when they felt uncomfortable, which could be a good lesson in intercultural communication for Japanese students. The interview was conducted in a conversational style in Japanese, and Chinese students A and B from the 2020 class were asked open-ended questions during a simultaneous online lecture for an hour.

The interview topics were based on the above-mentioned questions, which covered reasons for studying in Japan, personal and professional plans, unpleasant experiences in Japan, Chinese people's perceptions of Japan, and the Great Cultural Revolution (Appendix A). After a short introduction by the two students, the Japanese students voted online for the questions on the list, and the interviewer picked the ones with the most votes.

The Chinese students' videos were switched off according to their preferences, as were the other students, while the interviewer's video was on. During the interview, the interviewer, observing the Chinese students' responses, helped them understand the intent of the questions by rephrasing the original questions, giving them examples, and ensuring that the Japanese audience correctly heard their remarks.

Although most of the interview was a conversational interaction, when Chinese people's negative impressions of Japan came up as a topic, the interviewer gave a short lecture on Japanese people's affinity toward China, showing a graph (Figure 1): after normalization of diplomatic relations between Japan and China, a majority of respondents had affinities toward China in the 1970s; the percentage of people who "do not feel affinity" gradually increased after the Tiananmen Square Incident in 1989; and a majority "do not feel affinity" in the 2000s, reflecting chilled bilateral ties.

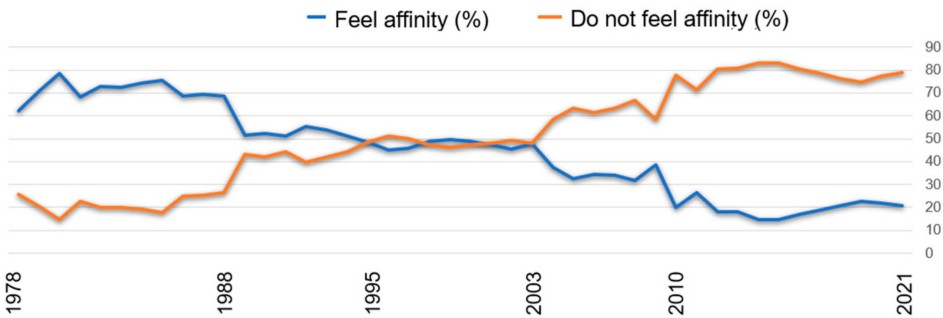

**Figure 1.** Japanese people's affinity toward China. Source: The Public Opinion Survey on Diplomacy by the Cabinet Office of Japan [2].

After class, the students submitted reflections on the interview and messages to Chinese students as an assignment. These comments were anonymized and shared with the class the following week. For example, some students referred to the graph, saying that "I know that the professor meant no harm, but she should not show [the graph to the Chinese students]" and that "I felt sad sensing the international students got hurt." The instructor appreciated their empathy and gave them the following feedback: the deteriorating bilateral relations had nothing to do with the international students themselves [37], and if they knew about it, they would not need to blame themselves when they suffered prejudice.

*2.4. Data Collection and Analysis*

As this study was concerned with students' perceptions, qualitative content analysis (QCA) was used to analyze the texts of their post-interview reflections. QCA is suitable for inductive category development based on research questions and theoretical background [34,35]. The data for the QCA were reflections submitted after the interview and students' comments on the course that were collected at the end of the semester. All data were obtained, analyzed in Japanese, and translated into English for this study.

To answer the research questions, the texts on what stimulated the Japanese students' positive remarks (RQ1) and the Japanese students' thoughts on stereotypes (RQ2) were extracted, which were from 90 students in total. The extracted segments were open-coded, and the resultant open-coded data were labeled as subcategories and more abstract categories. This process was repeated several times to finalize the categories. Multiple codes from the same students were counted as one to avoid duplicate entries.

**3. Results**

Although some students' questions collected before the interview showed biased views about Chinese people (e.g., selfish and dirty), their post-interview impressions were mostly positive or neutral, and there were no negative references to the Chinese students.

*3.1. Reasons for Japanese Students' Favorable Perceptions*

The coding processes generated three main categories of favorable perceptions: languages, manners, and empathy (Table 1). The most frequent codes were Japanese students' "amazement at the Chinese students' fluent Japanese" (the category of languages), their observation that the Chinese students "like Japan" and "think like grown-ups" (the category of manners), and their wishes for "more exchange opportunities" (the category of empathy).

**Table 1.** Reasons for Japanese students' favorable perceptions.

| Main Categories | Generic Categories | Subcategories | Codes (Number of Students) |
|---|---|---|---|
| Languages | Chinese students' fluency in Japanese | Appreciation of their good command of Japanese | - I am amazed at their fluent Japanese (19).<br>- They are fluent just after a few years' learning (5).<br>- They are easy to understand (3).<br>- They have a better command of Japanese than I do (2). |
| | | Recognition of their efforts to learn Japanese | - I feel their efforts of learning Japanese (2).<br>- They tried their best to communicate (1).<br>- I want to teach them Japanese (1). |
| | Chinese students as a role model | Wishes to speak foreign languages | - I want to speak English as they do Japanese (3).<br>- I want to speak Chinese with them (1). |
| | | Expression of struggles in learning Chinese | - The Chinese language is difficult to learn (2). |
| Manners | Chinese students' positive attitudes | Good impression of Japan | - They like Japan (10).<br>- They have a good impression (7). |
| | | Mature behavior | - They think like grown-ups (10).<br>- I respect their future plan (3).<br>- They behave more like a Japanese person (1). |
| | Chinese students' sincerity | Honest views on sensitive issues | - They have real and frank opinions (5).<br>- They did not avoid sensitive topics (4). |
| | | Good personality | - They are nice and kind persons (4).<br>- They did not blame Japanese people for unpleasant experiences (2). |
| Empathy | Impact on Japanese students' viewpoints | Renewed image of China | - My impression is now different from my negative impression (8).<br>- I felt closeness to China (2).<br>- I became interested in China (2). |
| | | Awareness of similarities between China and Japan | - I eat Japanese food when abroad as they eat Chinese food in Japan (5).<br>- We watch TikTok (4).<br>- We like Japanese anime (1). |
| | | Acceptance of cultural differences | - The importance of intercultural understanding (2).<br>- It is interesting to know cultural differences (2). |
| | Feeling of closeness | Confirmation of existing affinity toward China | - I had feelings of closeness (5).<br>- I am interested in Chinese entertainment and culture (3).<br>- It is the nearest Asian country (2).<br>- We had close economic ties (2).<br>- No good impressions but I feel familiarity (1). |
| | | Desires for friendship | - I want more exchange opportunities (12).<br>- I want to be friends (9).<br>- I will speak to my Chinese classmate (1). |
| | | Expression of moral support | - I wish them a successful stay in Japan (5).<br>- I pray for their success in future (3). |
| | | Inspiration for themselves | - I will work harder like them (2). |

### 3.1.1. Appreciation of the Chinese Students' Good Command of Japanese

This subcategory contained the codes with the highest frequencies among all subcategories. One of the typical reactions was, "their Japanese was very good". Some were amazed at their fluency "just after a few years' of learning Japanese" although "Japanese was difficult to master". Many seemed to recall their own difficulties communicating

in English after learning it for a long time. "Their Japanese is better than mine" is an often-used expression when a Japanese person praises a non-native speaker's Japanese.

### 3.1.2. Chinese Students' Good Impressions of Japan and Their Mature Behavior

Chinese students' attitudes toward Japan and their behaviors during the interview were also mentioned in the reflections. Japanese students were "glad to know that they had positive impressions about Japan" and happy "to hear them say they liked Japan many times". One of the students described that "they were more like Japanese than Japanese". This is another typical expression that describes foreigners as well-adapted to Japan.

Many students referred to the two interviewees' responses when the interview touched on sensitive and personal topics such as their uncomfortable experiences and views of history. They wrote, "it was respectable that they responded considering [the Japanese audience's] feeling and the atmosphere of the class" and "they had a mature way of thinking when they attributed a Japanese person's mean behavior not to Japanese in general but to the individual". Japanese students felt that the Chinese students were sincere when they did not avoid talking about their "shameful history" and "the issues that they may hesitate to speak". Although some students "expected to be criticized about anti-China sentiments in Japan, the Chinese students did not mention about it at all" and they "thought two students were very kind".

### 3.1.3. Japanese Students' Desires for Friendship

Many students "wanted to have more exchange opportunities with international students" and "wanted to be friends with them and speak more". They were interested in "knowing more good things about China" and "what was popular among Chinese youth".

### 3.2. Japanese Students' Awareness of Stereotypes

Compared with favorable impressions, fewer entries regarding stereotypes were counted. Table 2 shows the two main categories associated with Japanese students' awareness of stereotypes: analyses and opinions. The former category included what the students observed and noticed while reflecting on the interview. Writings related to their opinions on future intercultural encounters were sorted into the latter category.

**Table 2.** Japanese Students' awareness of stereotypes.

| Main Categories | Generic Categories | Subcategories | Codes (Number of Students) |
|---|---|---|---|
| Analyses | Influence on their impressions | News items negatively reporting on China | - The news picks up negative issues (6).<br>- I have an impression of Chinese people from news items and SNS (5).<br>- Media coverage highlights confrontation (2). |
| | | History and news are not necessarily neutral | - Japanese history is written from Japanese viewpoints (1).<br>- Knowledge of history may create stereotypes (1). |
| | Chinese people's impressions of Japan | Assumption of anti-Japanese sentiments | - Chinese people had a negative image of Japan (5). |
| | | Chinese students' positive views | - They had positive impressions of Japan (3).<br>- I wonder what other Chinese people think of Japan (1). |
| | Japanese people's impressions of China | Personal impressions before the interview | - I had a negative impression (5).<br>- I had stereotypical views about Chinese people (5).<br>- I had a discriminatory view (2).<br>- A huge gap exists between rich and poor (2).<br>- I did not feel affinity (2). |
| | | Observation of others' impressions | - Japanese people have negative impressions (3).<br>- Japanese people's negative impressions come from stereotypes (3).<br>- I feel sorry for our biased views about China (2).<br>- My family does not have good impressions (1). |

| Main Categories | Generic Categories | Subcategories | Codes (Number of Students) |
|---|---|---|---|
| | Adjustment of position | Finding mutuality | - The lack of affinity is mutual (2).<br>- Negative impressions are not very common at individual levels (1). |
| | | Noticing individual differences | - Only some Chinese people apply to stereotypes (6).<br>- Not all Chinese people think badly of Japan (5). |
| | Experience with Chinese people | Observation of bad manners | - I saw the bad behaviors of the Chinese (2). |
| | | Direct and indirect contact | - I had friends from China (4).<br>- My family used to visit China (2). |
| Opinions | Importance of exchange | Contact improves mutual images | - Direct exchanges improve impressions (2).<br>- Real voices change images (2). |
| | | Realizing lack of contact | - I have had no direct contact (2). |
| | Need for a wider perspective | Should not judge by limited information | - I am against one-sided judgments (3).<br>- It is not good to judge the nation without knowing it well (2). |
| | | Shake off preconceptions | - I need to think from wider perspectives (3).<br>- I should look at people as individuals (2).<br>- I need to be aware that news items may create stereotypes (2). |

### 3.2.1. Reflecting Pre-Interview Negative Views about China

While some students felt an affinity toward China before the interview, others reflected that they had "negative impressions of China". Some others referred to specific examples that "[Chinese people] do not follow the rules and are aggressive", and they "rip off [original characters and copyrighted items] and buy up [goods in Japan when they visit]". Students' assumption that "Chinese people have strong anti-Japanese sentiments" also indicates their stereotypical view. However, such negative impressions do not necessarily exclude closeness: "My impression is not so good, but I also feel close to China" because of "economic ties" and "geographical closeness".

### 3.2.2. Media Coverage Tends to Focus on Negative Issues

As the major source of these negative impressions, students pointed to media coverage in unison: "Many Japanese form their impressions about China shacked by prejudices and scenes of Chinese people aired in the news"; "I thought I knew about China although I only watched news coverage without meeting them directly"; and "I had believed the news featuring Chinese tourists' bad manners in Japan".

### 3.2.3. Noticing Individual Differences

The Chinese students' responses in the interview somewhat overturned the Japanese students' stereotypical views. A student had thought "Chinese students were overwhelmingly energetic"; however, when she listened to the interview, she felt they were "calm and reserved, which was quite different from her expectation". Similarly, other students wrote on individual differences: "Not all the Chinese are aggressive"; "I realized people are people regardless of nationalities, and differences depend on individuals"; and "I wonder if [the bad behavior of the] Chinese people I saw might have been an extreme case". Frank and realistic reflections should also be noticed: "Although I cannot say that I do not have stereotypes about China, I do not have negative impressions for the two Chinese students at all" and "I would rather hear what the Chinese general public think of Japan because Chinese students seem to have good impressions for Japan in general".

### 3.2.4. Importance of Mutual Exchange

Online contact with Chinese students stimulated the Japanese students to renew their attitudes toward Chinese people and intercultural exchanges. They reflected on their own experiences that they "had judged by impressions as they lacked contact with Chinese

people", and suggested that "by having contact between people in Japan and China, Japanese negative impressions for China and its people may improve positively". This is not limited to the Japan–China exchange: "It is not good to form impressions depending only on information from news coverage", and they "became aware of the importance of thinking with wider perspectives and flexibility to overcome biased views".

## 4. Discussion

When responding to sensitive and personal questions, Chinese students' sincerity evoked empathy [5,6,10]. Watching the interview influenced Japanese students' positive intentions toward future contact and exchanges with international students [18].

The Japanese students in this study did not talk directly with the Chinese students in class; instead, they listened to an interaction between the Chinese students and the instructor (the interviewees' videos were off), which can be understood as indirect contact [17,18,21]. However, according to Kim and Wojcieszak, who regarded reading online postings written by minority persons themselves as direct online contact [11], watching interviews can be interpreted as direct contact. The students referred to the experience as a "direct exchange" and commented that they "heard international students directly", which confirmed that they considered it to be direct contact. Their perception of "directness" was strengthened by the interview focusing on Chinese students' personal experiences and stories [10]. Moreover, the voting process might have enhanced their sense of directness. Although the two Chinese students were not typical representations of the Chinese people as understood by most Japanese audiences [13], they recognized as they watched the interview that the Chinese students belonged to the same university community [12], which resulted in the Japanese students' positive reflections.

The following subsection discusses the results in light of the two research questions.

### 4.1. Chinese Students' "Japaneseness" Contributed to the Favorable Responses

RQ1 focused on Japanese students' favorable perceptions of Chinese students after the interview. The QCA revealed that Japanese students highly appreciated the Chinese students' "Japaneseness", that is, their closeness to the Japanese in terms of language and manner, shown in their Japanese fluency; having a positive view of Japan; and their calm and mature behavior. The Japanese students accepted and welcomed the Chinese students because they spoke the local language [22,23] and behaved like the Japanese students without showing the "aggressiveness" that some Japanese students expected; The results suggested that the language was the major source that contributed to Chinese students' "Japaneseness". Chinese students speaking polite Japanese enhanced their respectful attitudes and assimilation into Japanese society. As this interview was conducted in Japanese, the Japanese students were willing and able to sense the interviewees' messages.

The same sentiment can be observed in Japanese companies' pro-diversity recruitment; they look for international students who are fluent in Japanese, understand Japanese culture, and behave like Japanese people, emphasizing "Japaneseness" over ability [38]. Although exchanges with international students can be good intercultural opportunities for most students, which was the original focus of this study, it is necessary to be mindful that emphasizing international students' "Japaneseness" may encourage them to assimilate into society, which results in "hiding diversity in a group" [26] (p. 67). It may further enhance ethnocentric attitudes among Japanese students and create another stereotype of international students.

### 4.2. Noticing Media Influences on the Students' View

RQ2 focused on the Japanese students' awareness of stereotypes. The QCA revealed that many were unaware that the media strongly influenced their perceptions of China before the interview; they thought Chinese people "thought ill of Japanese", "spoke loudly", and were "self-centered". While this implies that it is difficult for parasocial contact to work when media reports focus on negative aspects about China, the recent expansion of

Chinese entertainment in Japan could change the attitudes of the youth positively in the future [19].

Although the previous lecture introduced stereotypes, the students did not seem to recognize their pre-interview perceptions as stereotypes. However, watching the interview made them aware of the disparities between their pre-interview beliefs and the Chinese students' remarks in the interview. In this sense, the two international students were not common representations of the Chinese population [13], which helped the Japanese students notice their bias. When Japanese students examined dissimilarities, they realized that often-negative stories in the media influenced their views. The interview heightened their awareness of the perceptual distortions caused by stereotypes.

### 4.3. Pedagogical Implications

It is worthwhile to provide Japanese students with more exchange opportunities with international students [4–7], particularly Chinese students, considering the current stagnant bilateral relations that have a negative impact on mutual impressions [2]. Mutual interaction can also facilitate favorable perceptions and stimulate awareness of their bias against each other after contact. Their positive exchanges may hinder them from forming negative impressions in the future [15].

Many international students in Japan can communicate in Japanese, which can benefit Japanese students who have difficulty speaking foreign languages and hesitate to communicate interculturally due to language anxiety. Through intercultural contact students can imagine the other group's position, which may develop their empathy [20].

While recognizing the advantage of classroom contact with international students, careful consideration is necessary so that such exchanges do not lead to other stereotypes, even if positive, about Japanese-speaking international students. Thus, university instructors must create successful intercultural opportunities that encourage students to embrace diversity and look beyond foreign students' "Japaneseness" when communicating in Japanese.

### 4.4. Limitations and Future Research

Although this study identified "Japaneseness" as closeness to the Japanese in terms of language, behavior, and attitudes, it failed to examine how other elements, if any, contribute to the students' perceptions of "Japaneseness". Future studies should adopt different research designs to explore this complex issue in-depth. Further discussion is necessary to determine whether Japanese students' preferences for "Japanese-like" international students might create new stereotypes rather than enhance the benefits of diversity.

In addition, the interviewer's positive reaction and facial expressions, the only visible onscreen object, might have influenced the Japanese students' mood, leaving them with favorable impressions as an extended or vicarious contact effect [17,18]. Nevertheless, the findings suggest that increasing Japanese students' opportunities to interact with international students could help them appreciate different perspectives and make them aware of their biases.

Finally, it should be noted that the Japanese students were in contact with only two Chinese students who had a similar background, and the analysis of this study was based on their self-reported data during a short period after the intervention [24,25]. Considering that many Chinese are studying and working in Japan, future research on exposure to a more diverse group of Chinese and international students is necessary. Nonetheless, the process of analysis and the findings of this study have practical implications for classroom intercultural exchanges.

## 5. Conclusions

Japanese students generally grow up in a homogeneous environment with minimal first-person exposure to people and cultures that differ from their own. Their lack of confidence in cross-cultural communication was remarkable internationally [31]. Most

undergraduate students in Japan are approximately 20 years old [39], and on-campus diversity at a women's university or a predominantly female/male university can be particularly challenging.

Although intergroup contact is considered to improve mutual relations, the effects of online contact on campus have not been explored in Japan. This study attempted to identify the factors influencing Japanese students' favorable post-contact perceptions of Chinese students and their awareness of stereotypes. The study confirmed that online audio contact where students could not see each other impacted their perceptions and revealed that the Chinese students' "Japaneseness" significantly influenced the Japanese students' favorable perceptions, and the online interview made them aware of their biased views. The results of this small-scale study highlight the benefits of universities facilitating students' intercultural online contact to foster an appreciation for diversity by exposing them to different people and cultures and disconfirming students' stereotypes. Intercultural experiences can help them mature into citizens who can respect minority perspectives.

It should be noted that this study focused on Japanese students' perspectives and was conducted from the majority's point of view. Exploring interviewees' perceptions will provide more practical suggestions for intercultural communication respecting diversity. The findings illustrate the need for future research and practice on how students with no experience in communicating with minority citizens overcome ethnocentric perspectives and embrace diversity.

**Funding:** This research received no external funding.

**Institutional Review Board Statement:** The study was conducted according to the Komawaza Women's University code of ethics for research involving human subjects based on the guidelines of the Declaration of Helsinki, and the protocol was approved by the university's ethics committee (approval numbers 2020-006 on 9 October 2020 and 2021-015 on 20 September 2021).

**Informed Consent Statement:** Informed consent was obtained from all participants involved in the study.

**Data Availability Statement:** The data presented in this study are available upon request from the corresponding author. These data are not publicly available because of privacy concerns.

**Conflicts of Interest:** The author declares no conflict of interest.

**Appendix A**

This section provides the questions and answers of two international students (A and B) during the interview that were often mentioned in Japanese students' reflections.

- Why did you choose to study in Japan, and what was your plan after graduation?

  A: I like Japanese food and anime. I would like to continue my studies in graduate school.
  B: I became interested in Japan, as my uncle often told me about Japan. I would also like to engage in businesses that link Japan to China.

- Did your impression about Japan change after you came to Japan?

  A: No. Japanese people keep rules and they are honest about anything.
  B: It did not change. More precisely, I was able to understand Japan from various angles.

- What do you think is more convenient in China?

  A: Electronic payments and orders using smartphones. I did not carry a wallet with me when in China.

- What is great about China?

  A: Food culture. Since China is a large country, we have different food items and delicious dishes. There are eight culinary traditions by region. Taste changes according to the season.

- We have seen news reports about the gap between the rich and poor in China. What do you think about this?

    B: I have not felt a gap that much, but I think there should be regional gaps as the country is big.

- Have you had any unpleasant experiences in Japan?

    B: Well . . . I felt some older person spoke ill of me, although I did not understand exactly what was told. However, each person has a different character in Japan and any other country, and this only happens with a certain individual. I like most Japanese people.

- What do Chinese people think about Japan?

    B: Chinese people who come to Japan for travel and studies come to Japan as they like. Many Chinese wish to visit Japan because they find good things about Japan on the Internet.

- (A student recently read a science fiction novel set in the Cultural Revolution.) Could Cultural Revolution history be a good theme for a novel?

    A: This is a shameful and embarrassing history. [The interviewer apologizes for touching on a delicate topic.] Every country has a sensitive history that is difficult to explain, and (Chinese) textbooks do not describe the Cultural Revolution in detail.

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
