# Peer review of "Intercultural Opportunities to Evoke Empathy toward Minority Citizens: Online Contact with Chinese International Students at a Japanese Women’s University"

_societies, doi:10.3390/soc13060132_

Round 1

Reviewer 1 Report

Your study is interesting but needs to be grounded more firmly on the extant literature, ensuring the depth and criticality are both stronger. It would be important to review more in-depth the contemporary research and make sure you integrate it when analysing the data. A good idea would be to include a distinct discussion section. 

Generally, the standard of English is good and only needs minor edits. 

Reviewer 2 Report

The author(s) present an original and comprehensive study that advances the understanding of factors that interfere with cross-cultural relations among international students in Japan, especially students from China.

Considering the specificity of the university context, it would be interesting for the author(s) to provide some sociodemographic and academic information of the 132 students involved in this study. And if possible, the author/s could also provide possible differences that these variables could reveal in the results analysis.

On the other hand, it seems essential to describe the two Chinese students who were interviewed, as well as to justify their choice/invitation to the interview.

Finally, the authors should make explicit the ethical issues of the research.

Reviewer 3 Report

-Original research concept;

-Scientific reliability (however, there is question about the number of Chinese students participating in the research);

Reviewer 4 Report

This is an interesting, well-developed and well-written paper, whose results expand previous knowledge. The aim is clear, the title is informative, and the conceptualisation is thoroughly presented. The variables are defined and measured appropriately and the study methods are valid and reliable. The results are clearly presented.

A few language adjustments are still necessary though, especially in the abstract, which IMO does not reflect the quality of the paper and should, therefore, be rewritten in a more objective and clear fashion. For example, in lines 6-7, something seems missing: "...fostering students comfortable with multicultural environments" does not make sense. Lines 12-15 and 18-20 are quite confusing as well.

In line 57, a brief explanation of what the Hate Speech Act is would also be useful. 

Congratulations to the author on a well-achieved work.

Only a few minor revisions are required.

Round 2

Reviewer 1 Report

I would suggest that the author(s) strengthen the discussion part as well. 

n/a